# Epidemiology of Sports-Related Concussion in Japanese University Soccer Players

**DOI:** 10.3390/brainsci14080827

**Published:** 2024-08-17

**Authors:** Hiroshi Fukushima, Yutaka Shigemori, Shunya Otsubo, Kyosuke Goto, Koki Terada, Muneyuki Tachihara, Tatsuma Kurosaki, Keita Yamaguchi, Nana Otsuka, Kentaro Masuda, Rino Tsurusaki, Masahiro Inui

**Affiliations:** 1National Institute of Technology, Kurume College, Fukuoka 830-8555, Japan; ffha29@gmail.com; 2Graduate School of Sports and Health Science, Fukuoka University, Fukuoka 814-0180, Japan; goto_411@yahoo.co.jp (K.G.); kokitrd21@gmail.com (K.T.); mune.tachihara@gmail.com (M.T.); gd210010@cis.fukuoka-u.ac.jp (K.Y.); jinsg-627tbs@outlook.jp (K.M.); 3Faculty of Sports Health Science, Fukuoka University, Fukuoka 814-0180, Japan; n.otsuka0419@fukuoka-u.ac.jp (N.O.); inuiken@adm.fukuoka-u.ac.jp (M.I.); 4Center for Education and Innovation, Sojo University, Kumamoto 860-0082, Japan; otsubo@m.sojo-u.ac.jp; 5Department of Rehabilitation, Fukuoka University Hospital, Fukuoka 814-0180, Japan; 6Department of Health Sports Communication, Faculty of Human Sociology, Kobe University of Welfare, Hyogo 679-2217, Japan; t-kurosaki@sw.kinwu.ac.jp; 7Faculty of Human Sciences, Kyushu Sangyo University, Fukuoka 813-0004, Japan

**Keywords:** soccer, sports-related concussion, epidemiology

## Abstract

In recent years, sports-related concussion (SRC) in soccer has been extensively researched worldwide. However, there have been no reports of large-scale SRC studies among soccer players in Japan. The purpose of this study is to investigate the epidemiology of SRC among university soccer players in Japan. This descriptive epidemiological study collected data on the history of SRC and details of SRC injuries during soccer. The participants were university male soccer players belonging to the Japan University Football Association. SRC rates were calculated per 1000 athlete-exposures (AEs). A total of 5953 students participated in this study. The SRC rate was 0.10/1000 AE during total activities. The SRC rate during competition (0.42/1000 AE) was higher than in practice (0.04/1000 AE). The most frequent mechanism of SRC was “head-to-head” (26.9%), followed by “head-to-ball” (24.2%). During competition, the most frequent mechanism was “head-to-head” (30.8%), followed by “head-to-ground” (23.8%), and “head-to-ball” (19.3%) followed, while in practice, it was “head-to-ball” (34.8%), followed by “head-to-ground” (23.8%), and “head-to-head” (17.2%). Thus, there was a difference in the mechanism of injury between competition and practice. In this study, among Japanese university soccer players, the SRC rate was to be approximately ten times higher in competition than in practice.

## 1. Introduction

In recent years, the prevention of sports-related concussion (SRC) in contact sports has become a major topic worldwide for the prevention of severe head injuries. It has been reported that SRC is the most common type of sports-related head injury, and that the risk of SRC increases 2–5.8-fold the second time it occurs, and that with repetition, the risk of SRC, the severity of symptoms, and the time to recovery, increases [1,2]. Among other problems, contact sports participants, after the influence of head trauma, are prone to intracranial lesions such as acute subdural hematoma, which can be a serious head injury. In particular, most sports-related fatalities are caused by acute subdural hematoma, which is associated with a high mortality rate of 30–50%. In the United States, it has been reported that the reduction in SRC in American football has resulted in a decrease in fatal accidents [3]. Therefore, to prevent sports-related head injuries that are likely to cause serious accidents, it is essential to first prevent SRC.

A sport in which many people around the world participate, football is also a contact sport, and SRC research and prevention awareness-raising have been conducted extensively in Europe and the United States. Many people in Japan also participate in football [4]. And the prevention of SRCs during competition and post-SRC treatment have been emphasized in Japanese soccer competitions. In March 2012, the Japan Football Association (JFA) published the “Guidelines for Concussion in the Japan Professional Football League (J-League)” on the Internet to provide safety management for head injuries during soccer competitions. In November 2014, the JFA recommended the use of the “Guidelines for Concussion in Football” (Guidelines) for all individuals involved in soccer competitions in Japan [5]. In 2021, in accordance with the International Football Association Board (IFAB) notification [6], the JFA decided on “replacement (without re-entry) in the event of concussion” to ensure the safety of professional players. 

However, as far as we have been able to determine, no significant large-scale epidemiological studies have been conducted on SRC during amateur soccer competitions in Japan. Given the significantly larger number of amateur soccer players compared to professionals, understanding the current state of concussions in this population is crucial for effective injury prevention strategies. By conducting a comprehensive study on concussions in amateur soccer, we can identify risk factors, develop targeted prevention programs, and ultimately improve player safety. 

In this study, we will clarify the actual situation of SRC experience and SRC injuries among amateur university soccer players in Japan.

## 2. Materials and Methods

The participants were amateur university male soccer players who were members of the Japan University Football Association (=JUFA) (349 teams, 18,692 players) in 2020. We asked JUFA to conduct the survey and send a written request to the representatives of each team. The survey asked about age, history of soccer competition, and history of SRC during soccer competition; players with a history of SRC were asked additional questions, such as the injury situation (competition or practice), time of injury, mechanism of injury, and symptoms at the time of injury. If there were any unanswered items in each questionnaire, they were excluded from the data. These responses were collected using an Internet-based questionnaire (Google Forms) to collect the data. This research was conducted after obtaining ethical approval (No. 20-07M1) from Fukuoka University. The purpose of conducting this survey was explained in writing to the JFA and consent was obtained. The subjects, who freely participated in this study also received a written explanation, and their written informed consent and assent were obtained. The privacy of the participants was carefully considered during and after the study. 

In the present study, SRC was defined according to the definition provided by the Consensus Statement on Concussion in Sport [7]. Competition and practice were defined according to the consensus reported by Fuller 2006, with competition between members of the same team, and the warm-ups and cool-downs during competition defined as practice [8]. An athlete-exposure (AE) was defined as 1 athlete participating in 1 practice or competition. The rate of SRC was calculated based on the data collected in this survey and the general number of soccer practice sessions and competitions in Japan. The incidence of SRC was calculated based on the data collected in this study and the general number of soccer practice sessions and competitions in Japan.

### Statistical Analysis

Data were analyzed to assess the injury (history) rate, incidence, and mechanism of SRC in athletes.

## 3. Results

This survey received responses from 243 universities (69.6%) and 5953 (31.8%) amateur male soccer players belonging to JUFA. The age was 19.9 ± 1.3 years (18.0–24.0 years) of age, and the mean history of soccer competition was 12.9 ± 2.8 years (1.0–21.0 years). Midfielder (MF) was the predominant position at 39.4% in this survey, followed by Defender (DF), Forward (FW), and Goalkeeper (GK). There is no bias in the positional characteristics of the respondents (Table 1).

### 3.1. Percentage of Players with a History of SRC by Position

A history of SRC was found in 1600 (26.9%) players, who had a total of 2274 times of SRCs. Only one-time SRC is 1111 (69.4%) players, 356 (22.3%) players had two times SRCs, and 133 (8.3%) players had three or more times SRCs. The percentages of players with a history of SRC by position are shown follow in order of occurrence: 39.3% (237/603) for GK, 28.0% (562/2009) for DF, 25.8% (257/2344) for FW, and 23.2% (544/2344) for MF. When DF and MF were divided into center and sides, there was a significant difference between center back (CB) (32.1% (316/985) and side back (SB) (24.0% (246/1024). There was no significant difference between central midfielder (CMF) (23.0% (270/1172)) and side midfielder (SMF) (23.4% (274/1172)) (Table 2). The most common symptoms after SRC were headache, dizziness, and nausea. 

### 3.2. SRC Incidence, and SRC Rate (1000 AEs) by Position

In this study, since the average number of years of competition and the total number of SRC injuries to date were known, the rate of SRC was calculated based on the general number of practice sessions and competitions in Japan (4 competitions and 20 practice sessions per month); the rate was 0.10/1000 AEs for all activities. The incidence during competition was 0.42/1000 AEs and during practice was 0.04/1000 AEs. The AEs (competition:practice) for each position were as follows: GK (0.62:0.08), DF (0.45:0.04), MF (0.35:0.03), FW (0.41:0.04) (Table 3).

### 3.3. Mechanisms of SRC

There were 1628 SRCs caused by contact with other players and 646 SRCs that occurred without contact with other players, accounting for 71.6% of all SRCs caused by contact. The most common mechanism of SRC was due to head-to-head contact (603 times, 26.5%), followed by head-to-ball (550 times, 24.2%), head-to-ground (541 times, 23.8%), head-to-lower limb (233 times, 10.2%), and head-to-upper limb (156 times, 6.8%) (Table 4).

### 3.4. Mechanisms of SRC in Competition and Practice

SRC caused by contact with others occurred 1207 times (77.5%) during competition and 421 times (58.6%) during practice. The most common mechanisms in competition were “head-to-head”, followed by “head-to-ground” and “head-to-ball”. In practice, the most common mechanism was “head-to-ball”, followed by “head-to-ground” and “head-to-head (Table 4).

### 3.5. Mechanisms of SRC by Position

In this study, the GK had the highest incidence of SRC. In comparison to other positions, GKs had significantly higher head-to-lower limb contact (30.6% vs. 6.4%), while their percentage of head-to-head contact (9.5% vs. 29.7%) and head-to-ball contact (17.5% vs. 25.4%) was significantly lower in comparison to other positions (Figure 1).

## 4. Discussion

In recent years, the importance of responding to SRC during competition has been emphasized, and, in contact sports, when SRC occurs during competition, the competition is stopped to prevent serious accidents. However, in actuality, in the FIFA World Cup Brazil (2014) and Russia (2018), the English Premier League (2020), and the Asian final qualifying round for men’s football for the Paris Olympics (2024), concussions occurred during competition, yet players who suffered concussions continued to play. New guides such as SCAT6 are out, but even in the world’s top-level professional competitions, the response to the occurrence of SRC is not appropriate. Similarly, in Japan’s professional J-League (2023) soccer league, there have been several cases of players continuing to play after an SRC. Therefore, it is easy to imagine that amateur players with an inadequate playing environment are not adequately treated for SRC. Against this background, the present study was conducted to clarify the status of SRC among university soccer players in Japan.

### 4.1. SRC Incidence, and SRC Rate (per 1000 AEs) in Japan

In previous studies, the rates of SRC in middle school to university soccer players have been reported to be 0.15–0.49 per all activities, 0.53–1.38 per competition, and 0.04–0.24 per practice in 1000 players [9,10,11,12,13,14,15]. Although SRCs occur to some extent in soccer as in other contact sports, there have been no reports of large-scale epidemiological studies of SRCs in soccer in Japan. 

The SRC Rate (/1000 Athlete-Exposure) in this study is 0.11/1000 AEs in total activity, 0.41/1000 AEs in competition, and 0.04/1000 AEs in practice. In Europe and the United States, measures have been taken to decrease the risk of SRC, such as limiting the number of times heading is practiced from adolescence, but no such measures have yet been taken in Japan. Therefore, it is likely that SRC is less recognized by Japanese armature soccer players in comparison to EU and US players. In the present study, it is assumed that the analysis included cases of unrecognized as “unaware SRCs” [16] or misdiagnosed SRCs. Thus, although the rate of SRC in this study was lower in comparison to previous studies, the incidence of SRC in Japanese university soccer players may be higher than in the present study.

### 4.2. Rate of SRC by Position

Some previous studies [17,18] have analyzed positions in four categories: GK, DF, MF, and FW. However, we consider that there is a difference in SRC between players who tend to play on the sides of the pitch and those who tend to play in the middle of the pitch; thus, for a more detailed analysis, we divided DF into CB and SB, and MF into CMF and SMF. The rate of players with at least one SRC by position was as follows: GK, 38.1% (231/606); CB, 31.5% (311/987); SB, 23.1% (238/1029); CMF, 22.7% (267/1178); SMF, 22.6% (266/413); 24.6% (247/1006). For MF, there was no significant difference in injury rate between CMF and SMF. However, among DF, the SRC rate of CB was significantly higher in comparison to SB. Among the players with SRC, 41.2% of CBs, 40.1% of CFs, and 39.8% of GKs had more than two times SRCs. As previous studies have reported, the SRC rate differed depending on the position, with more GKs [9,17] and DFs [18] having a history of SRC. This study also demonstrated that CFs had a high rate of multiple SRCs, in addition to GKs and CBs. The players in these positions often have many opportunities to play near the goal. One characteristic of soccer is that it is more difficult to score in comparison to other sports. For this reason, the area in front of the goal is more aggressive than other areas, and players in positions with more opportunities to play in the area in front of the goal are likely to have more SRCs and a history of multiple SRCs. Kilkendall’s report [19] of a higher incidence of SRCs in the penalty area confirms the findings of this study. As in previous studies, there were significant differences between GKs and other field players in the mechanism of SRCs [9,13]. Compared to other positions, GKs had less “head-to-head” contact, but more head-to-lower extremity contact. This result is likely due to the rule that GKs can handle the ball with their hands, unlike players in other positions in the sport of soccer. Because GKs can use their hands, there is less “head-to-head” contact during confrontations. The only time GKs may intentionally “head a ball” is during an offensive set play at the end of a match when their team is losing, and the likelihood of “head-to-head” contact is extremely unlikely. The reason why the “head-to-head” contact rate for GKs is not 0% is that, in some cases, this occurs because, in some instances, after the goalkeeper has secured the ball, other players cannot stop their momentum in time and end up colliding with the goalkeeper’s head. GKs showed more head-to-lower limb contact than any other position for the following reasons. Players in positions other than GK do not have head-lower limb contact unless they keep their head low, or their opponent raises their feet high. In contrast, goalkeepers often lay their bodies on the ground to save the ball. At that time, the head is unprotected, and it is common for the opponent’s lower limbs to contact the head. Therefore, GKs have significantly higher rates of “head-to-lower limb” contact in comparison to other positions.

#### 4.2.1. SRCs Incidence Situations in University Soccer Players

All SRC cases were compared between competition and practice. In this study, 1555 (68.4%) SRCs occurred during competition, and 703 (31.6%) SRCs by occurred during practice, which amounted to a significant difference. The SRC rate (1000 AEs) in competitions was approximately 10.8 times higher than that in practice. Previous studies reported that the rate of SRCs in competition was approximately 5–15 times higher than that during practice [9,11,13,14], and our results were consistent with these studies. Most previous studies analyzed the SRC caused by contact with another player. However, in the present study, contact with another player was divided into nine categories for a more detailed analysis (Table 4). In this survey, we found that in competition, 30.4% of SRCs occurred due to head-to-head contact, 24.0% occurred due to head-to-ground contact, and 19.4% occurred due to head-to-ball contact. In contrast, in practice, 34.7% of SRCs occurred due to head-to-ball contact, 24.1% occurred due to head-to-ground contact, and 16.6% occurred due to head-to-head contact. There was a significant difference in the rates of SRC due to head-to-head contact, head-to-ball, and head-to-lower extremity contact between competition and practice. We present below a background of the main causal risks of SRCs inferred from the results of this study.

#### 4.2.2. “Head-to-Head” Is the Main Cause of the SRC

Among head-to-head contact injuries, 79.9% occurred during competition, which is approximately four times more than during practice. And the incidence rate (per 1000 AEs) was approximately 20 times higher than that during practice. One of the reasons for this is that players behave differently in practice and competition. In fact, players gave answers such as “I avoid forcible contact during practice” to prevent head injuries in this survey. Emery et al. reported that competition is more aggressive and competitive than practice, and that more contact between players results in more injuries [20]. And the present study also showed that in practice, the players avoided unreasonable contact out of concern for their teammates and themselves. Thus, we are assumed to find a background factor of competitive environment that influenced the significantly higher frequency of head-to-head contact in competition.

#### 4.2.3. “Head-to-Ball” Is the Second Most Common Cause of the SRCs in Japan

In previous studies of competition among professional soccer players, “head-to-head” and “head-to-elbow” contact were the most common mechanisms of SRC, “followed by head-to-lower extremity”, “head-to-ground”, and others, with few SRC due to “head-to-ball” contact [21,22,23]. In contrast, previous studies of amateur middle school, high school, and university soccer players have also reported injuries due to contact with the “ball” or “equipment” [14,15]. However, these reports do not clearly state how SRCs occur during competition or during practice.

In our study, 19.4% of all SRCs in a match occurred due to “head-to-ball” contact (i.e., one in five SRCs occurred due to contact with the ball). During practice, the rate was 34.1% (i.e., one in three cases occurred due to contact with the ball). This result is due to Japan’s specific practice environment, where many players practice in a small area. In addition, the players avoided forcing contact with each other in practice, as such “head-to-head” contact was less frequent than in competition, and thus the “head-to-ball” contact rate was higher. The present study clarified the mechanisms of SRC during competition and practice, and we consider that the results more clearly reflect the current situation of amateur soccer players who do not have an appropriate playing environment. 

A limitation of this study is that it is not possible to ascertain whether respondents have an adequate understanding of SRCs. In cases such as “unaware SRCs” [16], where respondents are not aware, there may be more potential SRCs than in the present results.

The study elucidates that a high proportion of Japanese university football players have experienced SRC. There are significant differences between the professional and amateur playing environments in Japan. Although there have been advances in SRC diagnostics such as SCAT6 and diagnostic techniques, it is important to study the actual occurrence of SRC in the playing environment as well.

## 5. Conclusions

This article is the first large-scale epidemiological study conducted investigating SRC in Japanese university football players. In this study, the SRC rate was to be approximately ten times higher in competition than in practice. Among DF, the SRC rate of CB was significantly higher in comparison to SB. This study also demonstrated that CFs had a high rate of multiple SRCs, in addition to GKs and CBs. In this survey, we found that during competition, the most frequent mechanism was “head-to-head”, followed by “head-to-ground” and “head-to-ball”, while in practice, it was “head-to-ball”, followed by “head-to-ground” and “head-to-head”.

The SRC rate (/1000 Athlete-Exposure) in this study was 0.11/1000AE, which was less than in previous studies. At the time of this study, measures to reduce the risk of SRC, such as limiting the frequency of youth heading practice, had not been implemented in Japan, so concussion was not fully recognized, and there were cases of so-called “unaware SRC” [16] where respondents were not aware of SRC, which may have led to fewer results than in previous studies. A critical implication of this possibility is the necessity to enhance awareness of concussions among Japanese soccer players.

## Figures and Tables

**Figure 1 brainsci-14-00827-f001:**
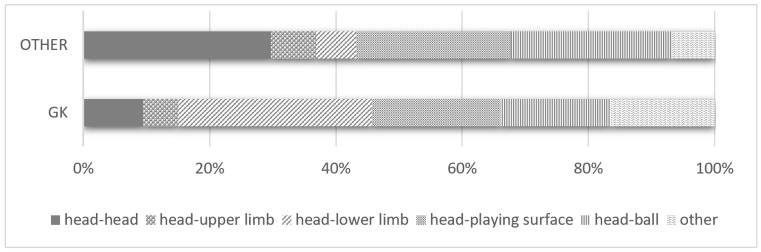
SRC Mechanism in Goalkeeper and other Positions.

**Table 1 brainsci-14-00827-t001:** Characteristics of Participants.

Position ^#^	Participants	Age (SD)	Play History ^¥^ (SD)
GK	603	19.93 (±1.22)	12.35 (±2.99)
DF	2009	19.99 (±1.26)	12.92 (±2.79)
MF	2344	19.89 (±1.26)	13.03 (±2.71)
FW	997	19.93 (±1.24)	12.86 (±3.03)
ALL	5953	19.94 (±1.25)	12.89 (±2.83)

^#^ Position: GK = Goalkeeper, DF = Defender, MF = Midfielder, FW = Forward. ^¥^ Play history: history of soccer competition.

**Table 2 brainsci-14-00827-t002:** Rate of players with a history of SRC by position.

Position ^#^	Total	SRC History (n) ^¥^	%	*p* Value	SRC Total ^Ψ^
GK	603	237	39.3%	-	359
DF	2009	562	28.0%	<0.001	801
MF	2344	544	23.2%	<0.001	741
FW	997	257	25.8%	<0.001	373
ALL	5953	1600	26.9%	-	2274
CB	985	316	32.1%	-	470
SB	1024	246	24.0%	<0.001	331
CMF	1172	270	23.0%	-	369
SMF	1172	274	23.4%	0.845	372

^#^ Position: GK = Goalkeeper, DF = Defender, MF = Midfielder, FW = Forward CB = center back, SB = side back, CMF = central midfielder, SMF = side midfielder. ^¥^ SRC history: Number of players with a history of SRC. ^Ψ^ SRC total: Total number of SRC.

**Table 3 brainsci-14-00827-t003:** SRC rate (1000AE) by position.

Position ^#^	Competition	Practice	Overall	Competition/Practice ^¥^
GK	0.62	0.08	0.17	7.75
DF	0.45	0.04	0.11	11.25
MF	0.35	0.03	0.08	11.67
FW	0.41	0.04	0.10	10.25
total	0.42	0.04	0.10	10.50

^#^ Position: GK = Goalkeeper, DF = Defender, MF = Midfielder, FW = Forward. ^¥^ The value obtained by dividing the rate of occurrence of competition by the rate of occurrence of practice for each injury mechanism.

**Table 4 brainsci-14-00827-t004:** Mechanisms of SRC in competition and practice.

	Competition	% ^#^	Practice	% ^#^	Overall	% ^#^	Competition/Practice ^¥^	*p* Value
head-head	479	30.8%	124	17.2%	603	26.5%	1.79	**<0.001**
head-upper limb	109	7.0%	47	6.5%	156	6.9%	1.07	0.678
head-lower limb	178	11.4%	55	7.6%	233	10.2%	1.50	**<0.001**
^Ψ^ head-playing surface	370	23.8%	171	23.8%	541	23.8%	1.00	0.995
head-ball	300	19.3%	250	34.8%	550	24.2%	0.55	**<0.001**
others	119	7.7%	72	10.0%	191	8.4%	0.76	0.059
overall	1555	100.0%	719	100.0%	2274	100.0%	1.00	**<0.001**
contact	1207	77.6%	421	58.6%	1628	71.6%	-	
no-contact	348	22.4%	298	41.4%	646	28.4%	-	

^#^ Percentages indicate the percentage of each injury mechanism when the total number of injuries is set at 100%. ^¥^ The value obtained by dividing the rate of occurrence of competition by the rate of occurrence of practice for each injury mechanism. ^Ψ^ head-playing surface: Contact with others before the head hits the playing surface.

## Data Availability

The original contributions of this study are detailed in the article. For further inquiries, please contact the first author or the corresponding author.

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
