# Peer review of "Epidemiology of Sports-Related Concussion in Japanese University Soccer Players"

_brainsci, 2024, doi:10.3390/brainsci14080827_

Round 1

Reviewer 1 Report

Comments and Suggestions for Authors

Introduction- overall well organized, describes relevant background to the study (e.g., soccer is now the sport with highest participation in Japan). One addition I feel would be helpful would be to more explicitly state why a large epidemiological study on SRC during amateur soccer in Japan is needed and what value it could have.

Methods- Well-described. Is it possible to include the survey itself as supplementary material? I feel presenting the number of unanswered questions that were excluded from the data (lines 75-76) would be helpful.

Results/Discussion/Conclusions-

One finding that stood out most to me- the incidence of SRC in the present study is lower than previously reported figures (section 4.1 in discussion). I would highlight this more in the conclusion, including your consideration that this may represent an undercount of the true incidence. One important takeaway from this possibility is that recognition of concussion in soccer players in Japan may need to be improved, which I believe is important to highlight more in the conclusion.

Another possibility I wonder about regarding this point is that you later make reference to survey answers regarding approach to competitions/practices- for instance, lines 241-243 note that one answer in the survey suggests that some players avoid contact during practice. Could there be other explanations for why SC incidence could be lower in soccer in Japan, based on playing style? If this is something that could be backed up by these survey questions/answers, then I would recommend additionally including these survey results in the manuscript.

There also are some concepts introduced in these sections that I believe would benefit from additional focus earlier in the manuscript to establish these concepts:

-Lines 122-124 note that the general monthly scheduled includes four competitions and 20 practices. I would include this definition earlier in the manuscript, in the methods section. Furthermore, I would recommend expanding on this- for instance, is this something that is mandated by JUFA? Is there any possibility that there is variability in the number of competitions and practices? (if so, would include this as a limitation in discussion)

Lines 259-260 make note that many players in Japan practice in smaller areas- this is another area I would recommend expanding on. Can you define what the standard competition field dimensions are and how the smaller areas differ in comparison? Are there any reports of how common smaller practice areas are? 

Comments on the Quality of English Language

As a primary English speaker, I felt that the quality of the English language in this manuscript is generally good and easy to follow. 

Author Response

Introduction- overall well organized, describes relevant background to the study (e.g., soccer is now the sport with highest participation in Japan). One addition I feel would be helpful would be to more explicitly state why a large epidemiological study on SRC during amateur soccer in Japan is needed and what value it could have.

As you point out, I explained why a large-scale epidemiological study on SRC in Japanese amateur soccer is needed and the value it could have.

Methods- Well-described. Is it possible to include the survey itself as supplementary material? I feel presenting the number of unanswered questions that were excluded from the data (lines 75-76) would be helpful.

I will submit the survey form as a supplemental document, as you have indicated.

Results/Discussion/Conclusions-

One finding that stood out most to me- the incidence of SRC in the present study is lower than previously reported figures (section 4.1 in discussion). I would highlight this more in the conclusion, including your consideration that this may represent an undercount of the true incidence.

In the conclusion, I have emphatically indicated that the incidence of SRC is low.One important takeaway from this possibility is that recognition of concussion in soccer players in Japan may need to be improved, which I believe is important to highlight more in the conclusion.

Thank you for pointing out this important perspective. However, I forgot to add it to Revision on this system and will revise and submit it as soon as possible.

Another possibility I wonder about regarding this point is that you later make reference to survey answers regarding approach to competitions/practices- for instance, lines 241-243 note that one answer in the survey suggests that some players avoid contact during practice. Could there be other explanations for why SC incidence could be lower in soccer in Japan, based on playing style? If this is something that could be backed up by these survey questions/answers, then I would recommend additionally including these survey results in the manuscript.

This perspective is also very interesting. However, we are still in the process of formally reporting the results.

There also are some concepts introduced in these sections that I believe would benefit from additional focus earlier in the manuscript to establish these concepts:

-Lines 122-124 note that the general monthly scheduled includes four competitions and 20 practices. I would include this definition earlier in the manuscript, in the methods section. Furthermore, I would recommend expanding on this- for instance, is this something that is mandated by JUFA? Is there any possibility that there is variability in the number of competitions and practices? (if so, would include this as a limitation in discussion)

The general number of practices is not required by JUFA, and the number of competitions and practices varies from college to college. This was additionally described.

Lines 259-260 make note that many players in Japan practice in smaller areas- this is another area I would recommend expanding on. Can you define what the standard competition field dimensions are and how the smaller areas differ in comparison? Are there any reports of how common smaller practice areas are? 

I have looked for prior research on this point but could not find any.
I am a long-time soccer person, I know the current situation where more than 100 people practice on a single soccer court, and I am not aware of any Western practice that exceeds 40 people on a single soccer court from what I have seen and heard.
Should I delete this sentence as well since I could not find any prior studies or formal reports?

Once again, thank you for taking time out of your busy schedule to give us many pointers for our paper.

Reviewer 2 Report

Comments and Suggestions for Authors

Dear all,

Thank you for the opportunity to review this manuscript. The manuscript aligns with the goal of the Brain Sciences, and the topic offers information for researchers, professionals, and the Neurosurgery and Neuroanatomy section. However, like with prior evaluations on the epidemiology of sports-related concussions, I believe the information provided might need to be clarified. Consequently, some points are listed below:

Title

I suggest one of the following:

Epidemiology of sports-related concussion in Japanese university soccer players

1. Introduction

The introduction should make an excellent first impression. The introduction to the current manuscript is confusing. The introduction could be written in a more well-organized and flowing way.

Line 51: ‘SCAT 6 etc.’: it isn’t clear what you mean.

2. Materials and Methods

In line 69: The participants were amateur university male soccer players: please could you explain why you didn’t involve female players?

In line 70: Japan University Football Association (=JUFA) (349 teams): as far as I know, the participating teams of the JUFA are 24 teams (Sapporo, Tokai Sapporo, Sendai, Ryutsu Keizai, Komazawa, Meiji, Hosei, Waseda, Tsukuba, Kokushikan, Niigata, Tokai Gakuen, Chukyo, Nagoya Keizai, Kwansei Gakuin, Kyoto Sangyo, Biwako Seikei Sport College, Hannan, International Pacific, Takamatsu, Kochi, National Institute of Fitness and Sports in Kanoya, Fukuoka, Miyazaki Sangyo-keiei). Could you explain what the number ‘349 teams’ refers to?

In line 72: ‘The survey asked about age, history of soccer competition’: this data is not presented.

In line 75: ‘symptoms at the time of injury’: also, no data was presented for this question.

In the questionnaire, did you provide the participant with the definition of what a concussion is?

According to an epidemiological standpoint, the research study would be more interesting and valuable if it included additional types of traumatic brain and cervical injury besides concussion, hospitalization rates, post-injury deficits, interventions, follow-up, symptoms linked with concussion post-injury, and/or whether concussion education was offered.

Lines 93-95: Statistical analysis: The statistical methods aren’t sufficient enough, and in general it doesn’t make sense for interpretation of the data.

Results

All Tables can be constructed and displayed in a more organised and clear way.

Line 104: ‘Table 1. Characteristics of Participants’: actually, isn’t characteristics, it is presenting the age of players. What do you mean by the term ‘play history’, needs to be clarified? Please, present the data as (mean ± SD).

In Table 1: All the participants involved in the study with almost the same age (19 years) with the same SD (≈1.2). Are all participants in the Universities Football J-League Championship at the same university level? Isn't there diversity among the different university levels within each university?

Line 105: ‘SRC history’: not found in the table, the table is presenting only ages and play history.

In Table 1. the abbreviations GK, DF, MF, and FW need to be defined under the table.

Table 2 also isn’t clear. Columns 5 and 7 without heading.

In table 2. the abbreviations GK, DF, MF, FW, C, SB, CMF, and SMF need to be defined under the table.

Table 3. What is C/P?. is it Competition/Practice? I tried to look at the numbers, but they didn't show up anything for me.

Table 4. Is it possible to make the title clearer without abbreviations? Could you change the heading numbers 1, 2, and 3 by symbols (e.g., #, Ψ, ¥). Inside the table, row 4, ‘*3head-playing’: what is (*) refer to?

In Figure 1. SRC Mechanism in GK and other Positions: the figure's graphic grey colours are very similar. Please, fill with patterns (e.g., strips, dotted, spheres, grids, etc).

References

Please use the ACS style guide to be compatible with Brain Sciences’ guidelines. The ACS style guide is recommended, please follow.

https://www.mdpi.com/journal/brainsci/instructions

Best wishes,

Author Response

Title

I suggest one of the following:

Epidemiology of sports-related concussion in Japanese university soccer players

I agree with this comment.  Therefore  I Changed the Title.

  1. Introduction

The introduction should make an excellent first impression. The introduction to the current manuscript is confusing. The introduction could be written in a more well-organized and flowing way.

Line 51: ‘SCAT 6 etc.’: it isn’t clear what you mean.

The introduction was not as flowing as you indicated.
Therefore, I have removed unnecessary sentences and provided a clear explanation of the value of doing this study.

  1. Materials and Methods

In line 69: The participants were amateur university male soccer players: please could you explain why you didn’t involve female players?

This study is the beginning of a large-scale concussion study in Japanese soccer, and was conducted on male soccer players, a large population of competitive soccer players. After this study is completed, we would like to modify the issues found and conduct a study of female soccer players as well.

In line 70: Japan University Football Association (=JUFA) (349 teams): as far as I know, the participating teams of the JUFA are 24 teams (Sapporo, Tokai Sapporo, Sendai, Ryutsu Keizai, Komazawa, Meiji, Hosei, Waseda, Tsukuba, Kokushikan, Niigata, Tokai Gakuen, Chukyo, Nagoya Keizai, Kwansei Gakuin, Kyoto Sangyo, Biwako Seikei Sport College, Hannan, International Pacific, Takamatsu, Kochi, National Institute of Fitness and Sports in Kanoya, Fukuoka, Miyazaki Sangyo-keiei). Could you explain what the number ‘349 teams’ refers to?

The 24 universities you listed were participating in the national competition sponsored by JUFA, and at the time of the survey, there were 349 universities affiliated with JUFA.

In line 72: ‘The survey asked about age, history of soccer competition’: this data is not presented.

Age and history of soccer competition were presented in the results and Table 1.

In line 75: ‘symptoms at the time of injury’: also, no data was presented for this question.

The typical symptoms are shown in order of prevalence in line 119.

In the questionnaire, did you provide the participant with the definition of what a concussion is?

According to an epidemiological standpoint, the research study would be more interesting and valuable if it included additional types of traumatic brain and cervical injury besides concussion, hospitalization rates, post-injury deficits, interventions, follow-up, symptoms linked with concussion post-injury, and/or whether concussion education was offered.

I did not provide participants with a definition of concussion in this survey. I think the study would have been more valuable if it had included head injuries other than concussions, as you mention, and hospitalization rates. I will keep this in mind when conducting future research.

Lines 93-95: Statistical analysis: The statistical methods aren’t sufficient enough, and in general it doesn’t make sense for interpretation of the data.

The sentence regarding statistical methods has been removed.

Results

All Tables can be constructed and displayed in a more organised and clear way.

Line 104: ‘Table 1. Characteristics of Participants’: actually, isn’t characteristics, it is presenting the age of players. What do you mean by the term ‘play history’, needs to be clarified? Please, present the data as (mean ± SD).

In Table 1: All the participants involved in the study with almost the same age (≈19 years) with the same SD (≈1.2). Are all participants in the Universities Football J-League Championship at the same university level? Isn't there diversity among the different university levels within each university?

Line 105: ‘SRC history’: not found in the table, the table is presenting only ages and play history.

In Table 1. the abbreviations GK, DF, MF, and FW need to be defined under the table.

Table 2 also isn’t clear. Columns 5 and 7 without heading.

In table 2. the abbreviations GK, DF, MF, FW, C, SB, CMF, and SMF need to be defined under the table.

Table 3. What is C/P?. is it Competition/Practice? I tried to look at the numbers, but they didn't show up anything for me.

Table 4. Is it possible to make the title clearer without abbreviations? Could you change the heading numbers 1, 2, and 3 by symbols (e.g., #, Ψ, ¥). Inside the table, row 4, ‘*3head-playing’: what is (*) refer to?

In Figure 1. SRC Mechanism in GK and other Positions: the figure's graphic grey colours are very similar. Please, fill with patterns (e.g., strips, dotted, spheres, grids, etc).

I have corrected all tables and figures according to what you had pointed out.

References

Please use the ACS style guide to be compatible with Brain Sciences’ guidelines. The ACS style guide is recommended, please follow.

https://www.mdpi.com/journal/brainsci/instructions

I have modified it according to the ACS Style Guide as recommended.

Once again, thank you for taking time out of your busy schedule to give us many pointers for our paper.

Reviewer 3 Report

Comments and Suggestions for Authors The manuscript entitled Epidemiology of sports-related concussion in all university soccer players in Japan, 2020 presented from Hiroshi Fukushima et al., is interesting however it seems a review or systematic but not an original article. It is not clear whether the authors analyzed video postmatch or they interviewed soccer players. the authors must explain this critical point. The authors must improve the results also including graphs and figures. How did they evaluate the different injury??? The introduction is really poor. The authors must improve this paragraph.

Author Response

The manuscript entitled Epidemiology of sports-related concussion in all university soccer players in Japan, 2020 presented from Hiroshi Fukushima et al., is interesting however it seems a review or systematic but not an original article. It is not clear whether the authors analyzed video postmatch or they interviewed soccer players. the authors must explain this critical point. The authors must improve the results also including graphs and figures. How did they evaluate the different injury??? The introduction is really poor. The authors must improve this paragraph.

I have made significant revisions to the introduction as you have suggested.
The research methodology was a questionnaire survey of college soccer players.
I have revised the entire text, including the figures and tables.

I apologize for the inconvenience, but I would appreciate it if you could review the paper again.

Round 2

Reviewer 2 Report

Comments and Suggestions for Authors

Dear all,

The authors provided acceptable responses to the comments and have amended the paper appropriately.

However, the references section is still not suitable with the journal guidelines. Please use the ACS style guide to be compatible with Brain Sciences’ guidelines. The ACS style guide is recommended, please follow.

https://www.mdpi.com/journal/brainsci/instructions

Regards,

Reviewer 3 Report

Comments and Suggestions for Authors

The authors improved the quality of the manuscript